# Energy Evolution: Forecasting the Development of Non-Conventional Renewable Energy Sources and Their Impact on the Conventional Electricity System

**Vadim A. Golubev** [1]![ID], **Viktoria A. Verbnikova** [2], **Ilia A. Lopyrev** [3], **Daria D. Voznesenskaya** [3], **Rashid N. Alimov** [2], **Olga V. Novikova** [4] **and Evgenii A. Konnikov** [3],*

1   Laboratory for Digital Modeling of Underground Oil and Gas Reservoirs and Well-Test-Analysis, Peter the Great St. Petersburg Polytechnic University, 195251 Saint Petersburg, Russia; doveva@mail.ru
2   Graduate School of High Voltage Energy, Peter the Great St. Petersburg Polytechnic University, 195251 Saint Petersburg, Russia; verbvika@mail.ru (V.A.V.); alimov.rn@edu.spbstu.ru (R.N.A.)
3   Graduate School of Industrial Economics, Peter the Great St. Petersburg Polytechnic University, 195251 Saint Petersburg, Russia; ilyalo1808@mail.ru (I.A.L.); vdaria0722@gmail.com (D.D.V.)
4   Graduate School of Nuclear and Thermal Power Engineering, Peter the Great St. Petersburg Polytechnic University, 195251 Saint Petersburg, Russia; novikova-olga1970@yandex.ru
*   Correspondence: konnikov.evgeniy@gmail.com; Tel.: +7-961-808-4582

**Abstract:** The development of the world's electric power systems goes back over a century. During this period, the overwhelming majority of states have formed stable, typically centralized systems for generation, transmission, and distribution of electrical energy. At the same time, technologies, primarily for energy generation, are steadily developing, which leads to the emergence of potentially effective technological solutions based on fundamentally new energy sources. The most rapidly expanding group at the moment are renewable energy sources (RES). This fact is due to the significant coverage of the potential environmental and economic benefits of using technologies based on RES in the information environment. At the same time, the process of transformation of traditional electric power systems, by integrating generation technologies based on the use of renewable energy sources, is extremely resource-intensive, and also potentially reducing the level of sustainability and efficiency of the entire system functioning as a whole. This thesis is primarily true for exclusively centralized power systems. The purpose of this study is to create a forecasting model for the development of non-conventional renewable energy sources (NCRES) for short, medium, and long term, which makes it possible to form an action plan to ensure a reliable and uninterrupted supplying of consumers, taking into account the existing electric power system. The developed model made it possible to identify the most promising directions of NCRES from the integration point of view, and for them the quantification and clustering of the information environment was carried out, which made it possible to identify key trends and the specifics of the development of technological solutions for these directions of renewable energy sources. The developed tool and systemic conclusions formulated on the basis of its application make it possible to develop mathematically sound solutions in the direction of managing the development of traditional electric power systems based on the integration of NCRES.

**Keywords:** renewable energy sources; non-conventional renewable energy sources; RES; NCRES; electric power system; information environment

## 1. Introduction

The world is currently on the verge of a fourth energy transition to the widespread use of renewable energy sources and the displacement of fossil fuels. The rate of these changes, the speed of transition, and the impact on the already established electricity system are associated with high uncertainty [1]. Any global changes in technologies require risk assessment within the framework of existing systems, since the sustainability and

continuity of energy supply determine not only energy security, but also the stability of economic development of individual regions and the country as a whole. Modern electric power systems in many countries have been formed for more than a century and their key elements are the fuel supply system, electric power generation, and the transmission and distribution systems of electrical energy. Modern trends encourage decentralization, on the one hand, stimulating the development of small generation facilities, including renewable energy sources, and, on the other hand, require taking into account the specifications of each technology being introduced [1,2]. The presence of significant negative consequences from underestimating the impact of an increase in the share of renewable energy sources in the balance of the power system's capacity is emphasized by the facts of massive disruption of power supply that consumers experience in different countries.

Non-conventional renewable energy sources (NCRES) include wind power; solar energy; small-scale hydropower; wave energy and energy of ebb and flow; geothermal energy; and bioenergy. Consider the dynamics of the development of renewable energy sources in different countries and in the world as a whole. This increment affects traditional energy systems which can cause significant issues, such as blackouts, due to instability of output for renewables. Such issues could influence the expansion of NCRES in some developing countries. Therefore, a model is required to predict the capacity of energy sources and predict possible problems within energy systems. Based on data provided by BP Statistical Review of World Energy [1] and the MGBM model described in the article, which is supposed to be the most accurate model [2], we made a trend model of the renewable's development (Figure 1). As is shown on a graph, there is a significant difference in the development of renewable energy sources in different regions of the world; however, there is a trend of increasing shares of renewables. On the downside, the model has a significant issue. It is a trend model which is based on statistical data, which leads to high errors during forecast periods in which rare events take place. For example, in 2014 there was a significant spike in renewable development, which can be explained by scientific exploration. Analyzing the 2014 spike [3] led us to conclusion that there were technology developments which led to a breakthrough and numerous projects in the renewables sphere. A trend model could not forecast these kinds of events, so our research is based on an originally developed mathematical model which includes developing projects and innovation trends within articles, so it makes prediction more valuable and accurate. Moreover, our model can be used for long-term forecasting because it is possible to evaluate future projects and some scientific trends, which makes our prediction more precise in long-term predictions, in contrast to traditional trend models which are influenced by cumulative error.

The purpose of this study is a predictive analysis of the prospects for the development of renewable energy sources, taking into account the existing energy system in the short, medium, and long term. In order to achieve this goal, it is necessary to accomplish the following tasks:

1. Investigate the specifics of the development of NCRES in relation to a universal object;
2. Propose a mathematical model that can be used to assess the development of NCRES and their mutual influence on the existing energy system;
3. Justify the features of the application of the forecasting model for the short, medium, and long term;
4. Investigate the degree of reliability of the application of the proposed model for certain technologies of NCRES;
5. Determine the most promising NCRES in the context of the identified predictive dynamics;
6. Identify technological trends typical for the most promising NCRES.

Non-conventional renewable energy sources world development forecast is shown in Figure 1.

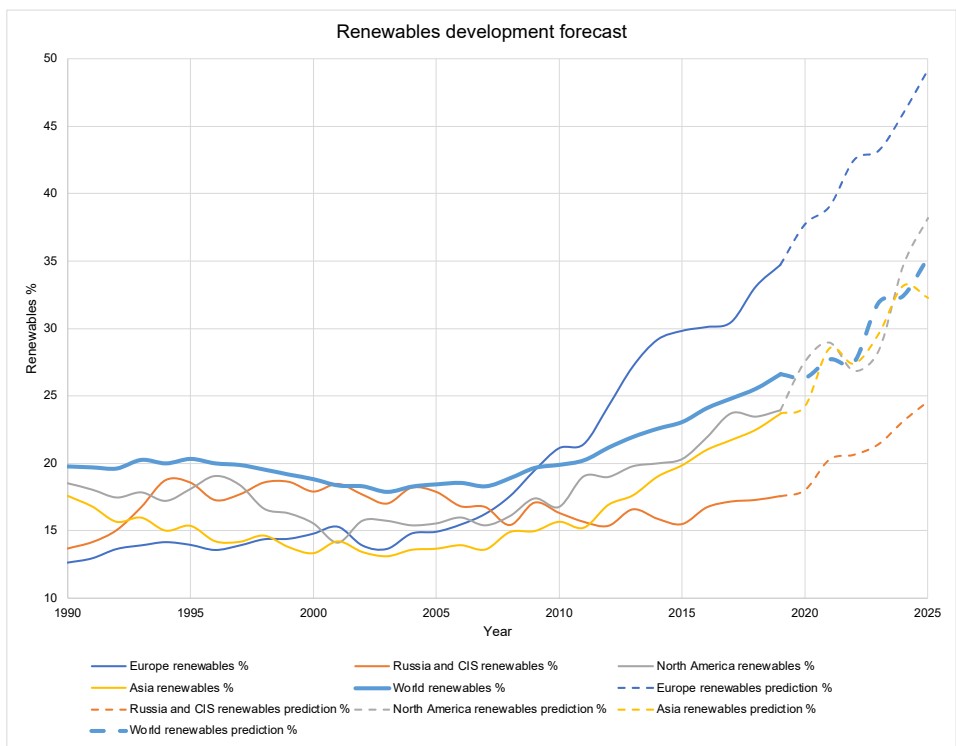

**Figure 1.** Non-conventional renewable energy sources world development forecast [1–3].

The established purpose of this study is not unique to the current scientific environment. This statement is due to the fact that the traditional energy system has been undergoing a rapid transition in the past 10 years with two most notable features, one of which is the high penetration of renewable energy generators using discontinuous renewable sources such as wind and solar. This transition is also causing a high degree of penetration of power electronic devices in generation, such as wind turbine converters and solar inverters, transmission, such as flexible AC or DC transmission system converters, and distribution/utilization systems, such as electric vehicles and microgrids. The development of modern power systems with multidirectional high penetration, i.e., with high penetration of renewable energy sources and power electronic devices, significantly affects the dynamics of the power systems and causes new sustainability problems [4].

At the same time, the importance of conventional power plants is increasing in parallel with the development of generation based on the penetration of renewable energy sources, its primary cause being the unsustainability of the generation process [5]. At the same time, a 10% increase in power generation using solar panels and wind could reduce the annual $CO_2$ emissions of the average thermal plant by about 4% [6]. Thus, researchers distinguish two key vectors of the impact of NCRES on the traditional power system: reduction of $CO_2$ emissions and increasing the level of unsustainability.

It should be noted that the electrical grid infrastructure is undergoing unprecedented transformations, mainly due to external factors: policies by government and regulatory authorities aimed at combating climate change; increasing opportunities and requirements of consumers; the development of distributed energy and the growth of electric vehicles quantity; the digitalization of grids with increasing integration of information and operational technologies; increased risks of cyberattacks; and energy market reforms, paving the way for completely new forms of competition. Attention should be paid to the fact that the impact of NCRES development is indirect. Transmission and distribution participants tend to interact more with each other in order to cope with the increased decentralized stochastic generation and changes in operating rules and procedures aimed at improving load regulation. These processes become the consequences of increasing the share of NCRES in traditional power systems [7].

In the confines of combating climate change, international experts generally consider only SPP (Solar Power Plant), WPP (Wind Power Plant) and some other renewable sources as an alternative to traditional sources [8]. At the same time, the prospects of using the traditional power system with reduced environmental impact and without the direct participation of NCRES should be noted. Technologies based on hydrogen and conversion of electricity into gas may become worldwide leaders [9,10]. Despite the prevailing trend toward electrification of energy sources and end-use, a balanced and reliable energy system is likely to require simple ways to transmit and store gas—possibly decarbonized gas. Consequently, energy-gas technology has enormous potential to provide synergistic coupling of sectors, which essentially means transmitting electricity for end-use in non-electric form. Gas infrastructure powered by greener gas can help ensure security of supply by bridging the mismatch between levels of peak power generation (most of the time by irregular renewable sources such as wind and solar) and demand [11].

Research of storage systems to enable the operation of NCRES in accordance with the mode of consumption is undoubtedly relevant [12,13]. Breakthroughs in electrochemical energy storage technologies—such as lithium or sodium ion batteries and supercapacitors—were used to create small-sized mobile electronic devices, medium-sized vehicles, portable and stationary devices, as well as for energy storage in large electric grids, paving the way to a new market with unlimited potential [14–16].

Energy recycling also has significant potential. Municipal solid waste-global production of 3.6 million tons per day gives an energy potential of 178 GW; hazardous waste—1.2 million tons per day, 43 GW; bio-waste—14 million tons per day, 685 GW; car tires—28,000 tons per day, 1.4 GW. The combined total is 907.4 GW, which compares to the entire U.S. installed capacity of 1100 GW [17]. As a different example, plasma recycling of municipal solid waste can provide about 5% of U.S. electricity needs. The most promising for the nearest future seems to be a complex approach which consists of waste management system development for each region [18,19].

NCRES has a variety of directions for development. In addition to the already common SPP technologies, chemical energy can be produced as a byproduct of utilizing sunlight [20]. Solar energy is a key element for many different ways to produce chemical fuel: production of biofuel from biomass; production of hydrogen in the process of microalgae; and production of biofuel by photocatalysis, performed by artificial devices [21–23]. Each of these methods has its own advantages and disadvantages. Successful development of these methods and overcoming the existing drawbacks requires further research in each of these sectors. Currently, the most popular solar fuel is biofuel derived from plant biomass. At the same time, molecular hydrogen is considered to be the fuel of the future, since it is a carbon-free chemical compound enriched with energy [24,25].

Thus, we can conclude that the prospects for the development of NCRES are extremely diverse, which is also multiplied by their level of impact on the traditional energy system. Parallel developing technological solutions, which cannot yet have a significant impact, but unambiguously contribute to the harmonization of the development of traditional energy together with renewable energy, are noted. However, the presented studies practically do not consider the problems of integrating generation systems based on renewable energy sources into existing traditional energy systems. Possible problems are extremely diverse and ignoring them can lead to both significant financial losses and the loss of energy security of entire regions. One of the most significant ways is to determine the most promising from the point of view of complex development of NCRES. It is necessary to effectively balance traditional and renewable sources. Only in such a case will the electric power industry develop towards minimal environmental risk and energy security [26]. Thus, the main gap in the current level of knowledge is the extreme inconsistency of research in the field of integration of modern energy generation technologies based on renewable energy sources into existing energy systems. The existing energy systems have been formed for decades and their current state and structure is determined both by the needs of energy consumers and by technological, natural, and infrastructural constraints of the environment.

Eliminating this gap requires systematization of the current trends in the development of technologies based on renewable energy sources and the formation the forecasts of development of this industry. This will make a significant contribution to the sustainability management of energy systems. This study raises the question of assessing the prospects of development and areas of impact of NCRES on the functioning of traditional power system, which can be largely investigated using statistical information processing. The scientific significance of this study and its difference from all those presented lies in a systematic approach to the analysis of the development directions of NCRES in the context of their integration with existing traditional energy systems. A systematic analysis of this issue is based on the analysis of objective statistical information and the results of quantification of the information environment, and does not use expert assessments, which increases the objectivity of the conclusions.

Thus, the main contribution of this research can be described by the following points:

1. Formation of a universal forecasting model for the development of basic NCRES technologies.
2. Identification of the most dynamically and steadily developing NCRES technologies.
3. Description of the specifics of the development of identified emerging NCRES technologies in the context of their integration into existing energy systems and increasing their sustainability.

This article, presenting the results of the current research, is divided into 5 main sections. The first section provides an overview of the current scientific basis and articulates a key gap in existing research. In the second section, the research methodology is presented in detail and the key methods for constructing a forecasting model for the development of the main NCRES technologies and an information environment analysis model describing the state of research of identified technologies are formed. The results section presents a forecasting model for the development of main NCRES technologies and identifies the most dynamically and steadily evolving NCRES technologies. The discussion section explores the information environment describing the development of the main NCRES technologies in the context of their integration into existing energy systems and identifies the key vectors for the development of renewable energy in general, taking into account the current scientific basis. Finally, the results are aggregated, and a brief summary of the study is presented.

## 2. Materials and Methods

The above-mentioned goal of the study determines the need for a consistent structural description and quantification of the development process of renewable energy in conjunction with traditional power systems. At the same time, the analysis of theoretical and methodological frameworks has established that the directions of NCRES development are extremely differentiated, both from a technological and economic point of view. Thus, the initial phase in this case should be the quantification and hierarchical classification of key renewable energy technologies, considering the dynamics of perspective development. However, in addition to the global technological development of NCRES, it is necessary to consider the importance of natural, territorial, and social factors that invariably affect the prospects for the use of the described technologies within a particular state or region. Consequently, the results of the initial stage should be specified taking into account these limitations. For the purpose of this study, the Russian Federation is chosen as a subject of analysis. There are several key reasons for this:

1. The Russian Federation is extremely vast from a territorial point of view, which determines a significant differentiation of natural and climatic conditions of NCRES-based projects development.
2. The Russian Federation has significant reserves of traditional energy resources, which, in turn, determined the formation of a developed traditional power system, that is totally centralized and sensitive to technological transformations.

3. The Russian Federation has a significant scientific and economic potential for the development of non-conventional renewable energy sources, which is being confirmed by the projects implemented in this area.

Thus, the primary stage of this study is aimed at identifying the most perspective NCRES influencing traditional energy system. The effect will be more significant in developing energy systems, which start energy transition to renewables, so it is possible to use the Russian energy system as an example to prove the model. However, the identified areas of development are extremely multidimensional; thus, they need to be clarified. For these purposes, it is proposed to conduct thematic clustering of the informational scientific environment focusing on the identified promising spheres, and to identify key areas of research in these segments. System analysis of links between the identified key areas of research will allow us to highlight the specifics of development of non-conventional renewable energy sources in the context of traditional power systems.

As part of the research, a mathematical model was developed, which was used to evaluate the development of renewable energy sources and their mutual influence on the existing power system of the Russian Federation. The analysis began with a review of historical data on the dynamics of capacity of non-conventional renewable energy sources in the Russian power system (Figure 2) [27].

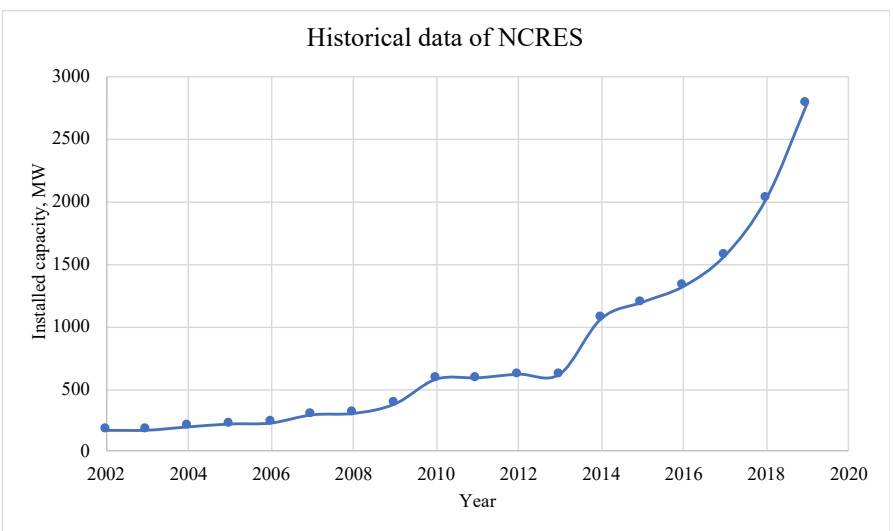

**Figure 2.** Historical data of NCRES development in Russia.

The graph has a characteristic non-linearity, which does not allow using simple polynomial time trends to assess the development of NCRES. Usage of such trend models can lead to a poor result shown in Table 1. Because of that we suggest developing a more advanced model which will be described below.

**Table 1.** Time-based trends in the development of renewable energy sources.

|  | R-Squared | MSE |
| --- | --- | --- |
| Linear trend | 0.56263715 | 308.4257 |
| Quadratic trend | 0.82746759 | 200.0688 |
| Cubic trend | 0.89081251 | 164.7449 |

None of the results obtained can be called satisfactory, because in the case of the cubic trend the number of variables becomes too large compared to the sample size, which makes this analysis irrelevant. On the other hand, the temporal influence on the development of the industry cannot be neglected.

In the analysis of technological development, inventions in non-conventional renewable energy technologies were analyzed, after which a correlation was found that all peaks in renewable energy growth are preceded by major discoveries within NCRES industries, as well as the completion of major projects. For example, in 2013, Sharp made a breakthrough in solar photovoltaic cells, increasing their efficiency to 44% [28]. In 2014, a project to build a biopower plant was implemented. In 2017, a large plant for the production of innovative solar panels "HEVEL" was built.

As for the current state of NCRES industries in Russia, solar energy is 0.72% of all energy capacity. Wind energy accounts for around 0.56% of the energy system. Other types of NCRES industries amount to about 0.3%. In its current state the influence is rather low, but will increase in the future development.

Due to the fact that non-conventional renewable energy is a knowledge-intensive industry, it is proposed to address the relationship between the creation of renewable energy facilities and the development of technologies of NCRES industries. However, in the development of technology there are a large number of innovations that do not receive any follow-up. In this regard, it is proposed to introduce a link between the development of technology and projects that have actually been implemented. A similar idea of linking the growth of emerging industries was expressed in the Thomas L. Heath series of books "The Thirteen Books of the Elements" [29].

Based on these judgments, it is suggested that the correlation between articles written on technology in any of the NCRES industries and actual projects using the technology described in the articles or using the result of the article in an actual project has to be evaluated. In addition, it is proposed to assess the general trend of projects within the selected NCRES industry through the value of the projects being developed. The last criterion will be the probability of discovery. This criterium is based on the correlation between the technologies of real projects and those described in highly significant articles, which can be considered a kind of fundamental work for the industry. Model scheme and results within every model step is presented in Figure 3.

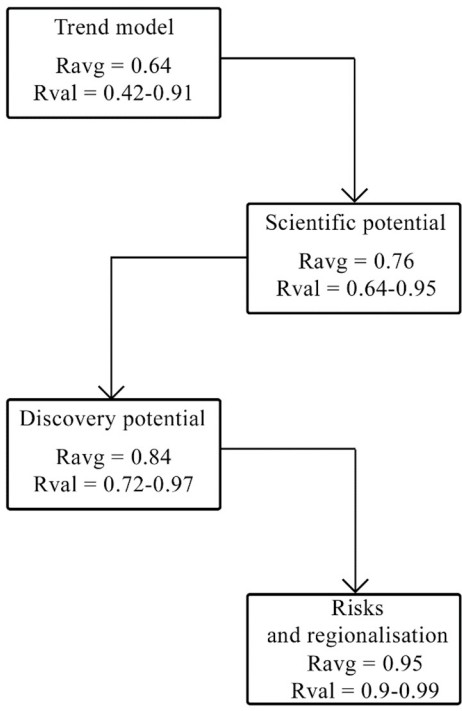

**Figure 3.** Scheme of mathematic model for prediction of NRCES development.

Ravg—Average R-squared value across all significant NCRES industries.

Rval—Is a range between lowest and highest R-squared value across all significant NCRES industries.

Based on the identified dependencies, three percentage values were calculated:

Estimated growth rate from developments-assuming an annual increase in the share in the balance of power of the industry, due to the projects implemented in it.

Scientific potential, showing the percentage of scientific activity that has real application within the industry.

Discovery potential, a percentage value reflecting the percentage of works of high importance with technologies that are implemented in projects of different companies.

As a final step, the model involves risks and regionalization which are supposed to make our prediction more precise.

For this purpose, algorithms were created in the Python 3 programming language for the article aggregators E-Library and Science Direct, as well as the procurement aggregators. With the help of the developed programs, articles and projects were uploaded after being sorted by keywords. Next, the technologies in the articles were analyzed and their correlation with the projects was confirmed. The Table 2 shows the results of the correlation analysis for 2020.

The result of the application of the described methodology is a model that allows us to identify the most promising dynamically developing types of NCRES. However, for the purposes of a predictive analysis of the prospects of renewable energy sources, considering the existing energy system in the short-term, it is necessary to describe the specifics of the most promising dynamically developing types of NCRES. To achieve this goal, it is necessary to analyze the information environment of each of the selected NCRES types. The automated algorithm for analyzing the information environment is shown in Figure 4.

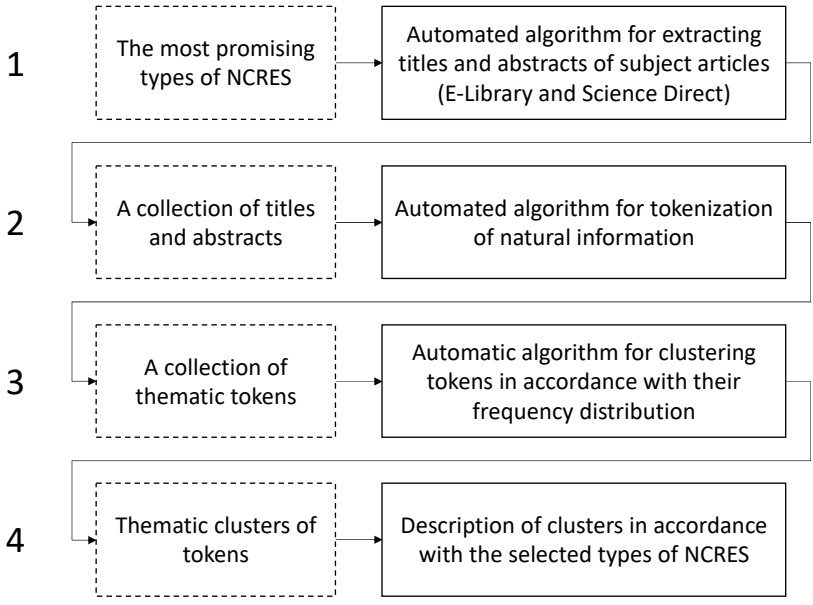

**Figure 4.** Information environment analysis algorithm.

The details of this methodology are presented in the study [30]. Based on the results of the application of this algorithm, a set of conclusions is formed regarding the specifics of the dynamically developing types of NCRES, which will make it possible to detail the specifics of the evolution of the energy sector. The complete algorithm of this study is shown in Figure 5.

**Table 2.** Calculated parameters for the model for 2020.

| Type of NCRES | Keywords | Company Purchases | Number of Articles | Estimated Growth Rate from Development | Scientific Potential | Discovery Potential | Highly Significant Points | Share in NCRES |
|---|---|---|---|---|---|---|---|---|
| WPP | WPP (ВЭУ-Ветроэнергоустановка) * | 1 | 1086 | 7.17% | 0.38% | 1.73% | 16 | 5.76% |
|  | WES (ВЭС-ветроэлектростанция) * | 34 | 666 |  |  |  | 5 |  |
| SPP | Solar panel | 22 | 2103 | 0.998% | 5.89% | 8.01% | 5 | 45.27% |
|  | Solar module | 4 | 1934 |  |  |  | 17 |  |
|  | Solar battery | 79 | 5488 |  |  |  | 20 |  |
|  | Charge controller | 2 | 382 |  |  |  | 0 |  |
|  | Inverter | 132 | 16996 |  |  |  | 55 |  |
|  | Batteries | Not relevant |  |  |  |  | - |  |
| SHPP (Small Hydropower Plant) | Hydroelectric turbine | 9 | 1978 | 21.29% | 0.64% | 0.91% | 6 | 5.22% |
|  | Hydroelectric generator | 2 | 755 |  |  |  | 3 |  |
|  | Small hydropower | 22 | 203 |  |  |  | 2 |  |
| GEOEP (Geopower Energy Plant) | Turbounit | 32 | 1206 | 7.52% | 0.26% | 0.66% | 8 | 4.46% |
| Tidal power plant | Tidal | 0 | 2045 | 0.00% | 0.45% | 3.55% | 43 | 0.06% |
| BIOEP (Biofuel Power Plant) | Bioreactor | 5 | 2891 | 0.04% | 0.63% | 2.39% | 29 | 30.91% |
| Waste | Waste incineration | 210 | 535 | 9.80% | 0.12% | 0.08% | 1 | 8.23% |
| Wave power plant | Wave stations | 0 | 438 | 0.00% | 0.10% | 0.25% | 3 | 0.00% |
| Landfill gas disposal | PSU—Pressure Steam Unit | 5 | 27 | 11.98% | 0.06% | 0.08% | 0 | 0.08% |
|  | Landfill gas | 2 | 232 |  |  |  | 1 |  |

* Research was made within Russian Science Citation Index, so keywords were used in Russian.

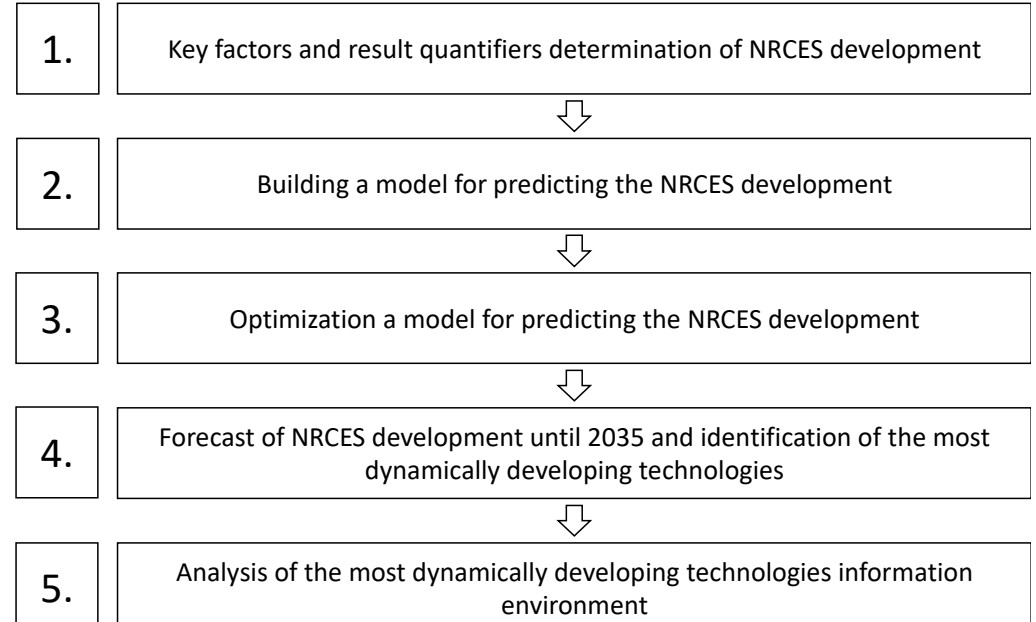

**Figure 5.** Research algorithm.

## 3. Results

Within the framework of this study, it was revealed that the main areas of development of non-conventional renewable energy sources are SPP, WPP, and BIOEP (Biofuel Power Plant). At the same time there is a significant increase in the number of SPP-based projects under development, which is again due to the recent development of technology in this area. Tidal power plants are promising in terms of development, but at the moment there are no implemented and functioning projects.

After the research, the linear trend model was extended by percentage coefficients. The final formula is as follows:

$$PO_i = PO_{i-1} \times \left(1 + \frac{K_r^{i-1}}{1.5}\right) + \left(1 + \frac{K_r^{i-1} + K_s^i}{100\%}\right) \times T_{tr} + P(V_{ex}) \times \left(1 + \frac{\sum_{n=0}^{i} K_s^n}{100\%}\right) \times T_{tr} \qquad (1)$$

where

1. $PO_i$—Industry volume in period i, expressed in installed capacity, in MW.
2. $T_{tr}$—Trend annual growth coefficient of the industries calculated on the basis of historical data on the development of the industry.
3. $K_r^{i-1}$—The estimated growth rate from developments in the previous period.
4. $K_s^i$—Current scientific potential.
5. $V_{ex}$—Discovery potential.
6. $P(V_{ex})$—Is a function returning 1 with probability $V_{ex}$ and returning 0 with probability $(1 - V_{ex})$.

Part of the calculations was the approximation that all projects in the procurements are implemented within 1.5 years of acceptance; this approximation was validated by the Pearson's Chi-squared criterion and the hypothesis that projects are implemented in 1.5 years with a 95% probability was accepted.

For any predictive model it is the predictive ability that is significant, which can be evaluated by calculating statistical parameters. To validate the model, we took data from 2002 up to 2020 with a lag of 2 years. The lag is caused by the fact that in order to use the linear regression algorithm a certain minimum sample is required, otherwise it will be impossible to construct a linear trend. In this regard, we took the data on the branches of NCRES and calculated the model. This dataset was taken as a validation to ensure that model is suitable for predictions within different states in energy evolution. In its

current state, renewables in Russia account for a small share of such energy sources in energy system, but its influence increases as the time goes by. Next, statistical indicators such as coefficient of determination, standard deviation, and coefficient of variation were estimated. The results are presented in the table below for some typical industries, in which the dependence is very different from the linear one. These industries had the highest deviations from the time trend, which prevented it from being used in the calculations (Table 3).

**Table 3.** Statistical parameters of the developed model.

| Parameter | WPP | SPP | BIOEP |
|---|---|---|---|
| R squared | 99.79% | 99.72% | 99.94% |
| MSE | 2.22 | 16.47 | 5.18 |
| Coefficient of variation | 4.16% | 5.67% | 1.84% |

As we can see, the coefficient of determination indicates that the model accurately describes the real dependence. The low values of the mean square error and the low coefficient of variation also indicate the high accuracy of the model. For the remaining industries, the coefficient of determination ranges from 98 to 99%, and the coefficients of variation do not exceed 7%, which indicates the versatility of this model for the non-conventional renewable energy industries. A visual comparison of the power NCRES graphs is presented below (Figures 6–8).

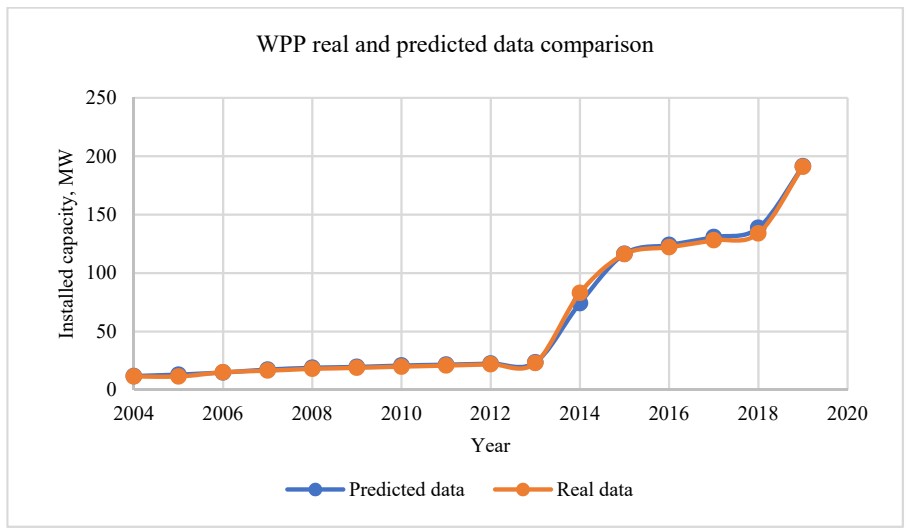

**Figure 6.** Comparative analysis of wind energy industry.

As we can see, the model has high accuracy, which makes it possible to make predictions on its basis and create certain patterns. It can be noted that for solar power plants prediction is in significance interval and it still can be used for a prediction modeling and forecasting, but has lowest accuracy across prediction of NRCES that is caused by significant non-linearity in data. This is due to the fact that a linear trend deriving from previous periods is used for each calculation. In turn, the linear trend becomes quite stable with a large sample, then the effect of outliers is not so great, in this case, the sample consists of five values, which are not enough for the trend to stabilize.

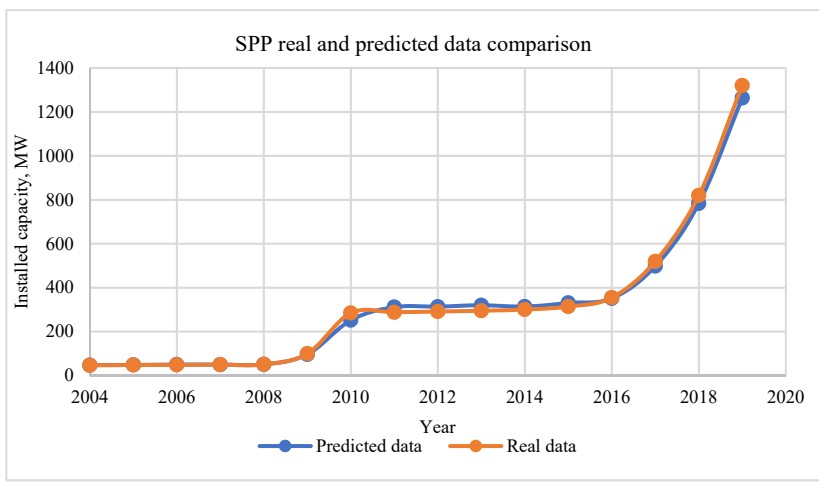

**Figure 7.** Comparative analysis of solar energy industry.

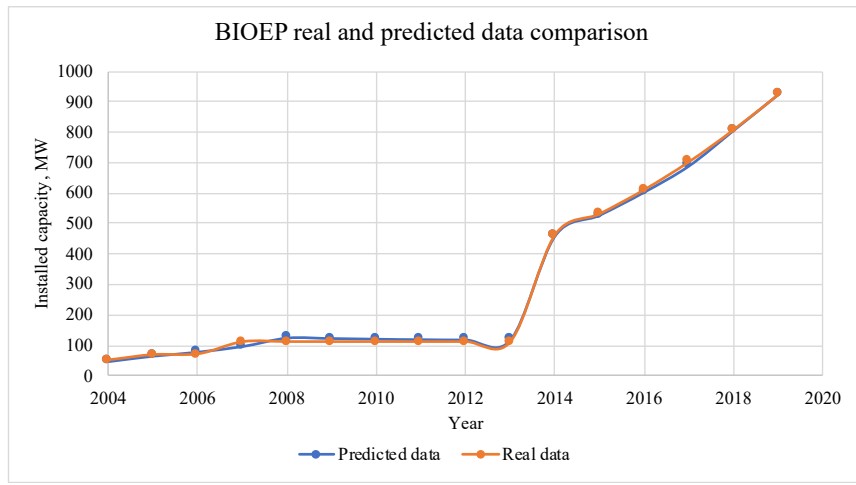

**Figure 8.** Comparative analysis of biological energy industry.

The presented model does not extensively highlight the specific sectors of renewable energy sources prevailing in each region of Russian Federation, focusing more on the rate of development of these technologies on the whole. This need derives from the inexpediency of placing certain types of renewable energy sources in certain regions due to their physical, economic, or other features. For example, all types of hydroelectric power plants need to be located in regions with an abundance of water resources due to the technical features of electricity generation. In this regard, regions with large volumes of water resources are more likely to build hydroelectric power plants. In order to take into account regional features in the model, the regions were ranked according to the development potential of the non-conventional renewable energy sectors [31,32]. For forecasting purposes, it is necessary to estimate the parameters of the model for future periods:

1. Scientific potential;
2. Potential of discovery;
3. Estimated growth rate of the industry from project development.

As part of data analysis on publications, a characteristic linearity was revealed in the percentage of publications whose technologies were applied in projects. On the basis of this, a linear trend was built and the scientific potential in the future period was calculated using the trend. The potential for discovery was estimated using the average percentage of high value articles that correlated with actual completed projects. From the point of

view of this indicator, it does not change over time, remaining rather low in any sector of non-conventional renewable energy sources.

The growth rate of the industry from projects in development is proposed to be estimated using the average value of the growth of industries over the recent years, since the influence of this indicator has a high impact in the short term. In the medium and long term, the key factor will be the scientific potential and the capacity for discovery, which allows us to use a rough estimate of this indicator due to the specificities of the model. Afterwards, based on the above conditions established for the model, the forecast was divided by time intervals into short-term (up to 5 years), medium-term (up to 10 years), and long-term (up to 15 years) forecasts. Due to the structure of the model, a certain segment of the formula presented above will affect the formation of each forecast. The first one is responsible for the short-term perspective, which does not take into account the potential of scientific discoveries, assuming that the development of the industry in this case is brought only by projects launched into implementation, since any innovative pilot project requires investment and is most often implemented in a much longer time frame. In the medium-term, for which the second item is responsible, scientific potential appears, showing that some designs that are in the pilot status can become full-fledged projects in the end and significantly affect the non-conventional renewable energy sector. In the long term, some discoveries are possible in each of the non-conventional renewable energy sectors, which was taken into account in the form of a probabilistic function of the opening potential. Within the framework of the baseline scenario for the development of non-conventional renewable energy sources, the probability had the pattern of a Gaussian distribution, and the number of cycles according to the Monte Carlo method was 1000 iterations. Figure 9 illustrates the rates of NCRES development over the years.

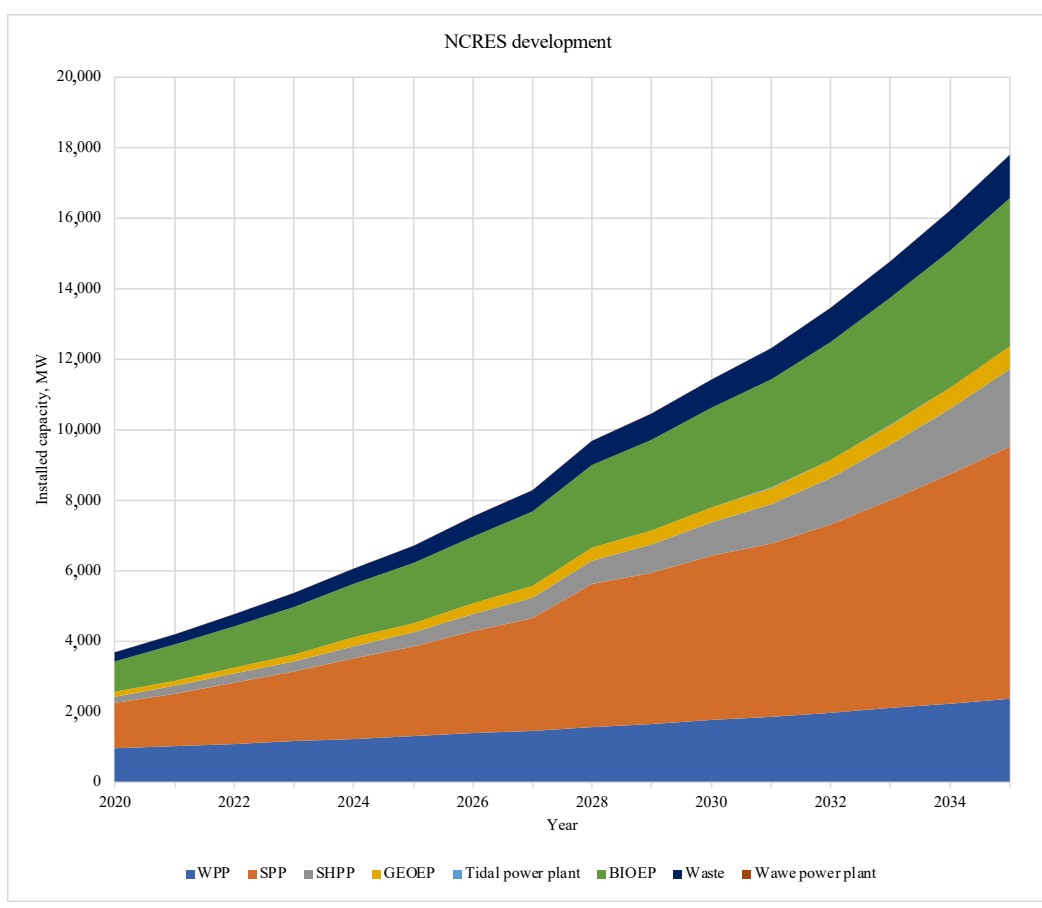

**Figure 9.** Forecast of RES development until 2035.

This graph reveals that some industries are developing more monotonously than others; for example, the SPP industry has a strong dependence on the emergence of new technologies. In our country, the industry has received extensive development, including scientific, which allows us to conclude the possibility of new discoveries in this sector of non-conventional renewable energy sources. Moreover, for all industries mentioned above as a whole, exponential development is typical, which may be a consequence of a favorable climate for the development of these areas in the long term, but this forecast is not completely accurate, since at the moment the risks of drastic structural change are not taken into account, which can seriously affect the further development of the industry. For forecasting purposes, it is not enough to rely solely on the current parameters of the system, since unforeseen events may occur in the future that may affect the development of a particular sector of non-conventional renewable energy sources. In this regard, it is necessary to introduce an additional risk correction parameter, estimated by PESTLE analysis using a fuzzy logic tool, which makes it possible to obtain a quantitative risk value, which will be taken into account in the presented model by shifting the distribution in the generator relative to normal, which seriously affects the change in the results after calculation by the Monte Carlo method. It should be noted that this study does not take into account the influence of solar cycles and global warming when predicting the development of solar and wind energy generation technologies.

Thus, it was revealed that the solar and wind energy industries are promising, as they have high potential for development at the moment. Within the SPP industry, a large development potential was identified, which allows us to expect high growth rates in the next 7–10 years. The industries of wave and tidal power plants will not become widespread due to the lack of large-scale projects being implemented, as well as new technologies that could help increase their efficiency. The growth rate of other industries (small hydroelectric power plants, geoPPs, bioelectric power plants, and the production of electricity from waste) will remain at an average level, due to their low level of development and the specificity of use of similar non-conventional renewable energy technologies. However, it is worth noting the systemic importance of BioEP, as a fundamental development vector from a technological point of view. The resource specificity of these technologies makes them universal, which largely determines their prospects. Therefore, it is necessary to study the information scientific environment of the solar, wind, and bioenergy industries.

## 4. Discussion

The conducted research allows us to assert that the dynamics of the development of existing NCRES in many respects confirms the conclusions of previous studies, and the primary driver of development is indeed environmental issues [6]. The conclusions of the study [8] were fully confirmed—the key technologies at the moment are solar and wind power plants. This fact indicates the extremely high importance of the climate change factor in the mediation of the development of renewable energy sources. Moreover, it is these technological areas that are currently being actively integrated into existing energy systems. At the same time, the dynamics of the development of technologies based on the use of hydrogen [9,10] or gas [11] is much less intense. The reason for this may be the logistic specifics of using these technologies, as well as the risk specifics. At the same time, only the analysis of the information environment will make it possible to establish the significance of the development of digital technologies within the framework of the development of NCRES [7].

In order to study the areas and topics related to non-conventional renewable energy sources, the most promising in the context of the study, it was decided to analyze the articles and research papers published over the past 3 years. Sciencedirect.com, an aggregator of scientific works, was chosen as the fundamental source of information. This site has several significant advantages over its analogue: only works from reputable journals and periodicals are published on the resource; it presents the results of research from around the world; and it is possible to fine-tune the scope of the search, which was used to simplify

the process of collecting and processing information. It is also necessary to note that articles aggregated within the framework of this source require mandatory detailed reviewing, which in turn does not exclude generalized and low-quality materials from the analytical array. It is also necessary to take into account the openness of these materials. This model was created to predict renewable capacity in an energy system, using a system theory to describe such a complex concept. Calculation of exact values for Russia's united energy system requires some simplifications in the data that are estimated. Citation rate was used to calculate science potential, because it represents the paper's popularity and relevance among other scientists. To account for the connection between articles and real-life projects, authors used correlation between technology in articles and real projects of companies that includes this technology. Citation rate was also used to calculate probability of discovery. However, in this case only the most valuable articles were considered (which have a citation rate of more than 50; value was calculated according to articles that were most influential in renewables in the past). To confirm the connection between new discoveries and their practical state, correlation between developed technologies and articles was used. This data were collected across the whole dataset and the model proved to be accurate according to series of statistical tests that were mentioned in 2nd block.

Since the key task of this stage of the research is the cluster analysis of tokens obtained during the processing of scientific papers on several main topics related to non-conventional renewable energy sources, a set of keywords should be chosen for each topic assessed. Search topics are selected according to their industry relevance. Consequently, tokens on the topics of solar, wind, and bioenergy will be researched. The fundamental difference between aggregators of scientific articles in comparison with search engines familiar to a modern person is that, due to the variety of articles, one list of keywords can get results from completely different areas of scientific research, which will be clearly demonstrated below. Accordingly, when choosing a set of keywords to form a sample of scientific articles, it requires accuracy in terminology and compliance with a strict limit on the number of selected words. The situation is also complicated by two additional factors: first, differences in the terminology of the Russian and English languages in the energy industry significantly limit the choice that allows you to maintain the accuracy of the query; secondly, when using automated methods of data processing, it is rather difficult to weed out scientific papers that do not correspond to the meaning, if their keywords match the request. As a result, the following keywords were chosen for data collection: for SPP—"solar power", for WPP—"wind power", for BioEP—"biogas energy".

As for the methods of processing the obtained data, for this work we chose an automated analysis of information using the Python 3 programming language, since, thanks to the almost unlimited possibilities of creating functions and programs, Python 3 allows you to create the most convenient analytic tools. This language was chosen because of its simplicity, the ability to run and test the program's performance in the process of writing code, as well as wide support from the community and the abundance of additional libraries that are perfect for analyzing both quantitative and qualitative information.

The program code has been divided into three parts for ease of editing and executing. The first part performs an automated entry to the site, selection of keywords, and loading the titles of all found articles into a separate *.xlsx file. For these purposes, in addition to the basic Python functions, the selenium, BeautifulSoup and time libraries were used. The selenium library, originally created to test the operation of browsers, today is often used for scraping—the process of automated collection and processing of data from the Internet. BeautifulSoup allows you to turn a page on the Internet into a set of text for the subsequent selection of the desired data by "tags"—elements of the code of a web page written in the html language. Time library—for setting delay timers for some program actions.

The next step was the transformation of the obtained data into a set of words for analyzing the frequency of mentioning various terms. For this, the following libraries were imported: nltk with different extensions—for tokenization (dividing text into words), lemmatization (highlighting the original form of a word to simplify aggregation and

analysis), and removing words without independent meaning (prepositions, conjunctions, articles)); the re library for extracting the title and individual words from the corresponding parts of the web page code; and pandas—for the formation of datasets and their subsequent study. Since language analysis requires fine tuning for automated functioning, several variables were introduced containing a list of designations for the key parts of speech and the choice of a list of stop words (words that have no independent meaning).

The last part of the code is the calculation of the frequency distribution of each token. For this, the collections library was used, which allows one, in particular, to count the total number of mentions of each word in the titles of articles. Thus, the final distribution file was generated.

The final stage of data processing before analysis was sorting the received tokens by frequency and excluding insignificant parts of the sample. For research purposes, those tokens are considered insignificant if the frequency of their mention is less than 13 times. Additionally, the tokens with the highest frequency of mention, as well as the keywords used for the search, are removed from the selection, since they have no meaning. Thus, for each area of data collection, a list of words was formed, with a volume of about 100 units, each of which was divided into clusters, according to the meaning of each of the tokens.

Based on the analysis results, the following conclusions can be drawn:

1. The most popular research topic related to all analyzed non-conventional renewable energy technologies is electricity generation and its technologies. For SPP, the corresponding tokens occupy about 6% of the sample (127 references), for WPP—about 2% of the sample, BioEP—4%. However, each of the industries has its own characteristics: articles on solar technologies place the greatest emphasis on photocatalytic decomposition as the main technology and works on wind energy place emphasis on a comparative analysis of existing technologies of wind power plants and solar power plants. In terms of bioenergy research, it mentions different ways to obtain fuels from biomaterials, accounting for 3.6% of the sample. Additionally, in the works analyzed, great emphasis is placed on the disclosure of different types of fuel resources or resources suitable for the mass production of biofuels. So, 6.7% (241 references) of the sample are tokens associated with this aspect.

2. Hydrogen energy is an integral part of almost all discussions related to the topic of non-conventional renewable energy sources. This fact is reflected in the research papers studied: each of the three samples contains references to hydrogen energy in varying degrees—from 1 to 1.5% of the sample. However, articles on bioenergy also contain descriptions of several technologies for hydrogen production: steam conversion, reforming, and gasification. Thus, it can be concluded that the discussion of bioenergy in the world scientific community is closely related to the production of hydrogen.

3. One of the important points in research on the topic of non-conventional renewable energy sources is also the ecological component of the relevant technologies. So, articles about SPP contain references to ecological tokens in the amount of 2% of the sample, WPP—3.2%, BioEP—an outstanding 5.4%, which is not surprising, given the lower environmental friendliness of bioenergy from the public point of view.

4. Another common theme for the three industries is accumulation systems, which occupy 2–3% of the sample for WPP, SPP, and BioEP. Based on this fact, it can be concluded that scientific community is highly interested in the issues of leveling and stabilizing the generation of non-conventional renewable energy sources using various storage devices, such as pumped storage power plants, battery complexes, and other technologies and resources, such as, for example, hydrogen, which was mentioned above.

5. It is also worth noting that scientific works on wind energy stand out from the rest because of the abundant interest in the issues of forecasting production. In particular, it is proposed to do this using neural networks, the mentions of which occupy about 1.5% of the sample. This trend was noticed only in this industry, which suggests the development of research directions, the results of which allow expanding the list of economically efficient wind generation regions in the future.

Thus, according to the results of this study, it can be concluded that the development process of the selected technological areas is extremely differentiated, which indicates fundamentally different technological development problems. The current gap in scientific knowledge is associated not with the lack of a unified position regarding the prospects for the development of renewable energy sources, but with the technological uniqueness of each of the selected areas. The technologies of solar and wind power plants are the most researched, which led to their dynamic development. At the moment, it is the technologies of solar and wind power plants that are developing in the context of practical use and, as a result, integration into existing energy systems.

## 5. Conclusions

Based on the conducted study using the tools of regression analysis, as well as quantification and clustering of the information environment, the features of the development of technological trends characteristic of the development of NCRES were identified. The main technologies that can influence the traditional electric power system are WPP, SPP, and bioenergy. The proposed mathematical model, with the help of which it is possible to assess the development of NCRES and its mutual influence on the existing energy system, considers the results of assessing the development of scientific potential and the potential for discovery. Since all the identified factors differ in time in terms of their impact, it is proposed to take into account the specificities of applying the forecasting model for the short, medium, and long term, namely projects at the stage of implementation, scientific potential, and the potential of discoveries. A high degree of application reliability of the proposed model for certain technologies of NCRES was demonstrated.

Based on the results of the study, models have been created for predicting the development of NCRES for the short, medium, and long term, which makes it possible to form an action plan to ensure a reliable and uninterrupted supply of consumers, taking into account the existing electric power system. This goal has not previously been considered separately using data on scientific potential and potential for discovery. The authors consider it important to note the versatility of the approach to forecasting the development of NCRES in a specific power system with the simultaneous planning of measures to ensure energy security. The obtained results contribute to the elimination of the current knowledge gap associated with non-systematic research in the field of integration of modern energy generation technologies based on renewable energy sources into existing energy systems and form the basis for further research in this area, potentially allowing us to significantly increase the sustainability of energy systems. As key limitations, it is worth noting that this study does not take into account the influence of solar cycles and global warming when predicting the development of solar and wind energy generation technologies, and also that the results of the analysis of the information environment can be interpreted in different ways. However, despite the existing limitations, the results obtained are of a multilevel theoretical and applied nature. The results can be used to develop plans for the development of the electric power system of the regions and the country as a whole.

**Author Contributions:** E.A.K. and V.A.G. designed the model and the computational framework and analyzed the data. V.A.G. and I.A.L. carried out the implementation. E.A.K. performed the calculations. V.A.V. and D.D.V. wrote the manuscript with input from all authors. O.V.N. and R.N.A. conceived the study and were in charge of overall direction and planning. All authors have read and agreed to the published version of the manuscript.

**Funding:** The research is partially funded by the Ministry of Science and Higher Education of the Russian Federation under the strategic academic leadership program 'Priority 2030' (Agreement 075-15-2021-1333 dated 30 September 2021).

**Acknowledgments:** The research was supported by the Peter the Great St. Petersburg Polytechnic University.

**Conflicts of Interest:** The authors declare no conflict of interest.

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
