# Peer review of "Energy Evolution: Forecasting the Development of Non-Conventional Renewable Energy Sources and Their Impact on the Conventional Electricity System"

_sustainability, doi:10.3390/su132212919_

Round 1

Reviewer 1 Report

Content comments, content evaluation:

Systemically, factually and with sufficient insight is described Introduction. Here, among other problems of renewable sources, the uneven transmission of generated electricity and the possibility of blackout due to excessive production (a typical problem of wind turbines) should have been mentioned. The very approach to the prediction of the development of renewable sources is also very beneficial. Despite the mentioned high potential of energy recycling, this is not mentioned in the model for Russia - it should be discussed.

Formal comments:

  • Research Methodology is only very briefly indicated, it needs to be elaborated. It should include research questions. In particular, it is crucial to describe in more detail the derivation of the model, including the simplifications that were adopted during its derivation. What theoretical methods were used - systems theory? Some parts of the methodology appear in the discussion - justification of the method of database selection, selection of programming language etc.
  • The description of resources in References needs to be adjusted according to the required format in Sustainability (many different bugs).
  • Author Contributions is missing.
  • minor inaccuracies: "which start enrgy transition to renewables,"
  • it is necessary to describe the pictures more precisely, namely: "Figure 2. Historical data of NCRES development." (in Russia); "Figure 3. Mathematic model scheme. (model of what?)
  • minimize / eliminate the use of formulations with "we"
  • Table 2. Calculated parameters for the model for 2020. - unify so that English is always listed first, then in parentheses the Russian keyword (part of the table can not have this order otherwise)
  • in some parts it seems that the reference to the source is given only after two sentences or even after the whole paragraph at its end (e.g. here: "This statement is due to the fact that the traditional energy system is undergoing a rapid transition in the past 10 years with two  most notable features, one of which is the high penetration of renewable energy generators using discontinuous renewable sources such as wind and solar. This transition is also causing a high degree of penetration of power electronic devices in generation, such as wind turbine converters and solar inverters, transmission, such as flexible AC or DC transmission system converters, and distribution/utilization systems, such as electric vehicles and microgrids. The development ofmodern power systems with multidirectional high penetration, i.e., with high penetration of renewable energy sources and power electronic devices, significantly affects the dynamics of the power systems and causes new sustainability problems [4].")
  • abbreviations should be consistently introduced (SPP)

Factual inaccuracies

"The main technologies that can influence the traditional electric power system are RES, SES and bioenergy." - RES refers collectively to renewable energy sources in the article, there should probably be "WPP" instead of "RES".

Author Response

Dear reviewer, thank you so much for your comments.

  1. This increment affects traditional energy system which can cause significant issues, such as blackouts, due to instability of output for renewables. Such issues could influence the expansion of NCRES in some developing countries. So model is meant to predict the capacity of energy sources and predict possible problems within energy system. To make the most precise model we make an overview of all technologies in NCRES sector, including recycling. Impact of recycling is rather high, but as a result of the article we presented results for 3 main sectors of NCRES, which doesn’t include recycling, because it’s impact is lower. 
  2. This model was created to predict renewable capacity in an energy system, using a system theory to describe such a complex concept. Calculation of exact values for Russia’s united energy system requires some simplifications in data that is estimated.  Citation rate was used to calculate science potential, because it represents the paper’s popularity and relevance among other scientists. To account for the connection between articles and real-life projects authors used correlation between technology in article and real projects of companies that includes this technology. Citation rate was also used to calculate probability of discovery. But in this case only most valuable articles were considered (which have citation rate more than 50, value was calculated according articles that influenced most in renewables in the past). To confirm connection between new discoveries and their practical state correlation between developed technologies and articles was used. This data was collected across whole dataset and model proved to be accurate according to series of statistical tests that were mentioned in 2nd block.

  3. Author contributions: Konnikov E.A. and Golubev V.A. designed the model and the computational framework and analyzed the data. Golubev V.A. and Lopyrev I.A. carried out the implementation. Konnikov E.A. performed the calculations. Verbnikova V.A. and Voznesenskaya D.D. wrote the manuscript with input from all authors. Novikova O.V and Alimov R.N. conceived the study and were in charge of overall direction and planning. Conflicts of Interest: The authors declare no conflict of interest.

  4. For that source reference is given exactly where it should be. But there were some issues across the article, so it was corrected. Thank you for your comment

Reviewer 2 Report

Although dependence on oil as an energy source will remain dominant for at least the next 25 years, there are remarkable policies which promote the diversification of energy sources. In this review, authors have developed a mathematical model to assess the development of non-conventional renewable energy sources and their mutual influence on the existing energy system. The review has been organized very well. The concluded results are interesting for the community and can be used to develop plans for the development of the electric power system of the regions and the country as a whole so target audience is broad. I would suggest few small changes.

  • It would be great if authors include the present status of non-conventional energy sources including solar, wind, bio and etc. in Russia
  • “CO2 emissions” needs to be corrected to the “CO2 emissions” through the text. “2” is superscript through the text that needs to be changed to subscript.

Author Response

Dear reviewer, thank you so much for your comments. 

  1. As for the current state of NCRES industries in Russia, solar energy is 0.72% of all energy capacity. Wind energy accounts for around 0.56% of the energy system. Other types on NCRES industries amount to about 0.3%. In its current state the influence is rather low, but will increase in the future development.

Reviewer 3 Report

I have reviewed the manuscript "Energy Evolution: Forecasting the Development of Non-Conventional Renewable Energy Sources and Their Impact on the Conventional Electricity System", Manuscript ID: sustainability-1385493. In this paper, the authors aim to develop a forecasting model for the development of non-conventional renewable energy sources (NCRES) for short, medium and long term in order to develop an action plan to ensure a reliable and uninterrupted supplying of consumers, taking into account the existing electric power system.

I consider that the article will benefit if the authors take into account the following remarks and address within the manuscript the signaled issues:

Remark 1: the main strong point of the manuscript consists in the fact that it approaches an interesting and actual topic for the experts in the field.

Remark 2: the main weak point of the manuscript under review consists in the explanations regarding the first year considered within the study, 2002. The authors should explain within the manuscript why they have chosen this year as first year of study, instead of a more recent one. In addition to this, I would like the authors to comment in the paper whether the data collected in 2002 are still relevant today, in 2021, in what concerns the same targeted parameters. The authors should provide explanations whether their study is consistent, whether the changes that may occur within the older dataset from the above-mentioned period and the current year risk altering the final result.

It will benefit the article if the authors take into account the accelerated pace at which inventions and discoveries in the field of information technology and Non-Conventional Renewable Energy Systems take place nowadays and provide in their study a series of insights towards the selected time moment, namely if the older periods of time are still relevant today and lead to reliable results, considering the vast range of existing Non-Conventional Renewable Energy Systems.

If the above-mentioned problems are solved, I consider that the paper will benefit if the authors address within the manuscript the following aspects:

Remark 3: the gap in the current state of knowledge. After having performed a critical survey of what has been done up to this point in the scientific literature, the authors should identify and state more clearly in the paper a gap in the current state of knowledge that needs to be filled, a gap that is being addressed by their manuscript. This gap must also be used afterwards by the authors, in the final part of the manuscript as well (when discussing the obtained results), where the authors should justify why their approach fills the identified gap in rapport with previous studies from the scientific literature.

Remark 4: aspects regarding the main contributions of the current paper. It will benefit the paper if in the final part of the "Introduction" section, the authors present the main contributions of their paper, eventually synthetized within a bulleted list.

Remark 5: presenting the structure of the paper. At the end of the Introduction, the authors should preview the structure of their paper, under the form: "The rest of the paper is structured as follows: Section 2 contains…". 

Remark 6: the flowchart. I consider that in addition to the actual explanations, in order to help the readers better understand the methodology of the conducted research, the authors should devise a flowchart within the "Materials and Methods" section, a flowchart that depicts the steps that the authors have processed in developing their research and most important of all, the final target. This flowchart will facilitate the understanding of the proposed approach and at the same time will make the article more interesting to the readers if used as a graphical abstract.

Remark 7: the research methodology. The authors have provided a series of details regarding the research methodology, as follows: At Lines 276-277: "For this purpose, algorithms were created in the Python programming language for the article aggregators E-Library and Science Direct.", and at Lines 428-435: "In order to study the areas and topics related to non-conventional renewable energy sources, the most promising in the context of the study, it was decided to analyze the articles and research papers published over the past 3 years. Sciencedirect.com, an aggregator of scientific works, was chosen as the fundamental source of information. This site has several significant advantages over its analogue: only works from reputable journals and periodicals are published on the resource; presents the results of research from around the world; it is possible to fine-tune the scope of the search, which was used to simplify the process of collecting and processing information." I do not contradict the value of this approach, or its relevance in this context, but I consider that the article under review will benefit if the authors explain in the paper what was the criteria in their research methodology based on which they have decided to analyze only papers from the two above-mentioned sources. In the current form of the manuscript, the methodology does not provide the rationale for the eligibility criteria, a complete rationale for the study design.

Remark 8: the "Results" section. At Line 393, the authors present Figure 8, entitled "Forecast of RES development until 2035", while afterwards at Line 412, they state: "solar and wind energy industries are promising". I consider that the authors should specify in the paper if, when devising the forecast and when declaring the statement from Line 412, they have taken into account the variability of solar radiation and wind speed, especially in the context of the Climate Change and Global Warming. In addition to these, as the forecasting period covers a long period of time (up to 2035) the authors should mention in the manuscript the impact of solar cycle and explosions on weather, on solar energy production and forecasting. According to the National Weather Service (https://www.weather.gov/), the actual solar cycle (entitled Solar cycle 25) began in December 2019 and, according to the calculations, the solar maximum will occur between 2023 and 2026, therefore within a period comprised within the forecasted period of the manuscript under review. Therefore, the authors should refer to this event and discuss if and how their research can take it into account.

Remark 9: discussing the obtained results - the comparison between the study from the manuscript and other ones is missing. After having presented the results, the authors should move forward and extend the comparison between their developed approach and results from the manuscript with other ones that have been developed and used in the literature for the same or related purposes. This comparison is mandatory in order to highlight the clear contribution that the authors have brought to the current state of knowledge. The authors should also highlight clearly what are the advantages and disadvantages when comparing their devised study with other studies from the scientific literature.

Remark 10: discussing the obtained results – insight. The paper will benefit if, after having discussed the obtained results, the authors make a step further, beyond their approach and provide an insight regarding what they consider to be, based on the obtained results, the most important, appropriate and concrete steps that all the involved parties should take in order to benefit from the results of the research conducted within the manuscript.

Remark 11: the "Conclusions" section of the manuscript. In this section the authors should avoid simply summarizing the aspects that they have already stated in the body of the manuscript. Instead, they should interpret their findings at a higher level of abstraction than in the previous sections of the manuscript. The authors should highlight whether, or to what extent they have managed to address the necessity identified within the "Introduction" section (the identified gap). The authors should avoid restating everything they did once again, but instead they should emphasize what their findings actually mean to the readers, therefore making the "Conclusions" section interesting and memorable to them. The authors should not restate what they have done or what the article does, they should focus instead on what they have discovered and most important on what their findings mean. The authors should also highlight current limitations of their study, and briefly mention some precise directions that they intend to follow in their future research work.

Remark 12: the software and the detailed hardware configuration. At Lines 453-454, the authors state: "As for the methods of processing the obtained data, for this work we chose an automated analysis of information using the Python programming language". It will benefit the paper if, along with the elements already presented, the authors specify details regarding the version numbers for the software and the detailed hardware configuration used to obtain the results.

Remark 13: the paper has been submitted to the Section "Energy Sustainability" of the MDPI Journal Sustainability. In this context, I consider that the authors should strengthen the main impact and relationship of their study and results with regard to Energy Sustainability. In the actual form of the paper, these connections are explicitly mentioned just a few times, and these mentions are not related to the results obtained in the manuscript under review, namely:

At Lines 29-32: "At the same time, the process of transformation of traditional electric power systems, by integrating generation technologies based on the use of renewable energy sources, is extremely resource-intensive, and also potentially reducing the level of sustainability and efficiency of the entire system functioning as a whole."

At Lines 51-53: "Any global changes in technologies require risk assessment within the framework of existing systems, since the sustainability and continuity of energy supply determines not only energy security"

At Lines 108-118: "The development of modern power systems with multidirectional high penetration, i.e., with high penetration of renewable energy sources and power electronic devices, significantly affects the dynamics of the power systems and causes new sustainability problems [4]. At the same time, the importance of conventional power plants is increasing in parallel with the development of generation based on the penetration of renewable energy sources, its primary cause being the unsustainability of the generation process [5]. At the same time, a 10% increase in power generation using solar panels and wind could reduce the annual CO2 emissions of the average thermal plant by about 4% [6]. Thus, researchers distinguish two key vectors of the impact of NCRES on the traditional power system: reduction of CO2 emissions and increasing the level of unsustainability. "

It will benefit the paper if the authors provide explicitly more details on this issue.

Other remarks.

Remark 14: information written in Russian language. A part of the information contained by the second column of Table 2 is in the Russian language. Sometimes, the information is provided in English and translated within brackets in Russian, while sometimes the information is provided in Russian and within brackets is translated in English. The authors should address this inconsistency.

Remark 15: Figures and Tables not referred. The authors should pay more attention to the details, as in the current form of the manuscript a series of errors has occurred. For example, Figures 5-8 and Tables 2-3 are not referred in the manuscript according to the recommendations from the Journal's Template.

Remark 16: run-on expressions. At Lines 473-474, the authors state: "prepositions, conjunctions, articles, etc.". In a scientific paper one should avoid using run-on expressions, such as "and so forth", "and so on", "and some other" or "etc.". Therefore, instead of "etc", the sentence should mention all the elements that are relevant to the manuscript.

Remark 17: the pages' numbering. On the second page of the paper, at the bottom right corner appears the page number: "2 of 17", and so on, up to the eighth page of the manuscript that is unnumbered and is followed by another unnumbered page, then by a page numbered as "2 of 17" and so on up to the last page of the manuscript that is numbered as "9 of 17". The authors should correct this inadvertence, the total number of pages is 17 and the pages numbering should be corrected.

Author Response

Dear reviewer, thank you so much for your comments. 

  1. We took data from 2002 up to 2020. This dataset was taken as a validation to ensure that model is suitable for predictions within different states in energy evolution. In its current state, renewables in Russia account for a small share of such energy sources in energy system, but its influence in-creases as the time goes by.

  2. We formulated a gap in the current level of knowledge and reformulated the conclusion based on its elimination.
  3. We have articulated elements of key contributions of our research and added them to the introduction.

  4. Remark 5 - Dear reviewer, thank you so much for your comment, we have completed this section and the introduction has become much more logical and structured.
  5. Remark 6 - Dear reviewer, thank you very much for your comment, we built this flowchart and added it to the "Materials and Methods" section.
  6. Remark 7 - Dear reviewer, thank you very much for your comment, we have explained in more detail the choice of these sources.
  7. Remark 8 - Dear reviewer, unfortunately, our methodology does not take into account the specifics of solar cycles and global warming. We agree that this is our oversight and thank you so much for pointing out this shortcoming. In subsequent studies, we will certainly take these aspects into account. Within the framework of this study, we pointed out this drawback.
  8. Remark 9 - Dear reviewer, we apologize for this misunderstanding, we forgot to include this block with the final version of the article. Thank you very much for pointing out this drawback, we have added this block.
  9. Remark 10 - We have formulated this conclusion.
  10. Remark 11 - Dear reviewer, we have tried to complete the conclusion.
  11. Remark 12 - Dear reviewer, we have indicated the version of the programming language. However, the deeper detailing of the technical specifics is extremely voluminous, and therefore we have added a link to the study, where this methodology is considered in detail.
  12. Remark 13 - Dear reviewer, we have tried to strengthen the presence of precisely the topic of sustainability of energy systems.
  13. Remark 14 - Dear reviewer, thank you very much, unfortunately we did not notice this technical error. We've adjusted everything.
  14. Remark 15-17 - Dear reviewer, we’ve made the correction.

Round 2

Reviewer 1 Report

Thank you for respecting my comments on the first version of the article.

Reviewer 3 Report

I have reviewed the revised version of the manuscript "Energy Evolution: Forecasting the Development of Non-Conventional Renewable Energy Sources and Their Impact on the Conventional Electricity System", Manuscript ID: sustainability-1385493 and I can state that the authors have improved it to a great extent.

This manuscript is a resubmission of an earlier submission. The following is a list of the peer review reports and author responses from that submission.

Round 1

Reviewer 1 Report

Comments to the Author(s):

In this manuscript, the authors proposed a forecasting model for the development of renewable energy sources for the short, medium and long term. The mathematical model is clearly presented, validated and drawbacks also are mentioned. Overall the research is interesting but the originality of the paper and comparison to the related research in this field should be more clearly and more strongly indicated in the Introduction section.  Please find the details below.

Other comments:

I. Comments to the Introduction section:

  1. The carefully current state of the research field is missing, the introduction should be rewritten and key publications should be cited. The authors stated that: "The purpose of the study is to create a forecasting model for the development of renewable energy sources for short, medium and long term,..." but for example state of art concerning mathematical models is missing.
  2. I suggest adding the reference to the sentence in lines 67-69. The reference to data presented in Figure 1 is needed.
  3. The acronyms should be explained at the first use (line 119- SES, WES).
  4. I suggest authors avoid lumping references, like in lines 136,144; for example [9-17]. It brings no information.
  5. The originality of the paper and comparison to the related research in this field should be more clearly and more strongly indicated in the Introduction section.

II. Comments to the Materials and Methods, Results and Discussion sections:

  1. I suggest rewriting the Materials and Methods, Results and Discussion sections according to the journal's Instructions for Authors. Currently, the information in these chapters is mixed. For example information about the algorithm (line 235) and mathematical model (line 256), the paragraph in line 411 or 420, etc. The current layout of the content may make the article difficult for the reader to understand.
  2. To better understanding, I suggest adding model parameter designations in Table 2.
  3. The article title suggests that RES impacts on the conventional electricity system will be calculated but the authors didn't present it.

III. Comments to the Reference section:

  1. References 1,2 should contain information about the journal name, vol., etc.

Author Response

Dear Reviewer, we are very grateful for your comments. In accordance with your comments, we have made the following adjustments to the article:

  1. We supplemented the introduction section and focused on its significance and differences from the reviewed studies.
  2. We really did not formulate the goal correctly in the previous version. Now we have reformulated the goal and now it is in line with the research results.
  3. We reviewed the current research of the area and correctly indicated all reviewed publications.
  4. We have added a link and explanations to Figure 1.
  5. We have added transcripts to all abbreviations.
  6. We tried to reformat the results presented in a more convenient perception for the reader. Unfortunately, it is not possible to add model parameters to the table due to their essential differentiation. Also, we would like to note that the problem of integration of NCRES and the existing traditional electric power systems is considered in detail in the final sections of the article and, the "Results" section.
  7. We have corrected references in the list.

We have tried to correct the article, focusing on the scientific significance of the study, and we very much hope for your favor.

Reviewer 2 Report

This paper addresses a forecasting model for the development of non-conventional renewable energy sources (NCRES) for short, medium and long term, in order to design an action plan to ensure a reliable and uninterrupted supplying of consumers, taking into account the existing electric power system. The developed model is used to identify the some promising directions of NCRES from the integration point of view and for them the quantification and clustering of the information environment. Based on the conducted study using simple tools of regression analysis, as well as quantification and clustering of the information environment, the features of the development of technological trends characteristic of the development of NCRES were identified.

This paper addresses a simple forecasting model without any scientific innovation.

Figure 1 represents, which is not sourced in the manuscript, provides the non-conventional renewable energy sources world development forecast. What is the difference, in scientific terms, the difference between the forecasting model proposed in this paper and the one used the produce figure 1?

Author Response

Dear Reviewer, we are very grateful for your comments. Our prediction model really uses the traditional regression methodology, but its content significantly differs from the existing models and the forecasting quality is statistically quite high. Also, we want to note that the scientific significance of the article is largely based not on the presented model, but on the conclusions and analysis of the information environment presented in the "Discussion" section. We have tried to reformulate the goal of the study and supplement the introduction section to make the scientific significance of our research more evident. Thank you so much for your question on Figure 1. Based on data provided by BP Statistical Review of World Energy and the MGBM model we made a trend model of the renewable’s development. As it’s shown on a graph there is a significant difference in the development of renewable energy sources in different regions of the world, however there is a trend of increasing share of renewables. On the downside the model is having a significant issue. It’s a trend model which is based on a statistic data, which leads to high errors during forecast period in case some rare events take place. For example, in 2014 it is a significant spike in renewable development, which can be explained with a scientific exploration. Analyzing 2014 spike led us to conclusion, that there were technology development which lead to a breakthrough and a numerous projects in renewables sphere. Trend model couldn’t forecast these kinds of events, so our research is based on originally developed mathematical model which includes developing projects and innovation trends within articles, so it makes prediction more valuable and accurate. Moreover, our model can be used for a long-term forecasting because it’s possible to evaluate future projects and some of scientific trends, that makes our prediction more precise in long-term in contrast of traditional trend model which is influenced by cumulative error. In other words, we still believe that the results of our research are significant, especially in the framework of the analysis of the information environment, and may be worthy of publication in a journal. We have tried to correct the article, focusing on the scientific significance of the study, and we very much hope for your favor.

Reviewer 3 Report

47-69 There are no citations included for your statements in the introduction, Include suitable citation for all the claims

Figure 1: It gives the world development forecast of non-conventional renewable energy sources, but there is on citation where these data are pulled from. Need to make sure these data are from reliable sources

67 - The description for Figure 1 is not in detail. Try to explain in detail

91-92 Again there is no proper citation or data analysis for such a strong statement

102, 105 - CO2  (2 should be in subscript)

119 - Give definition of SES and WES. I believe its solar and wind.

155 - NCRES are described as diverse, but the what's the focus of this paper is not clearly explained in the introduction part. Please redefine the last paragraph

177 - Russian federation is chosen for this study, so it cannot be applied globally? The title should be energy evolution in Russian federation. The current title is deceiving.

197 - Give more details regarding the mathematical model used

206 - Is there any reason behind time trend models used here ?

251 - Define BIOEP

297 - Any reason for the inaccuracy of the solar power data? And what are the uncertainties for all the data set ? How its predicted ?

302 - If there is a serious drawback on the lack of regional division in Russian Federation, then why this is chosen for the study ? This should be considered.

312-314 How are these factors shortlisted ?

Figure 6 - Describe the terms inside the figure, its hard to understand them without any description

353 - Why structural changes are not taken into account ? There is a lack of details in many places. Need clear flow

Author Response

Dear Reviewer, we are very grateful for your comments. In accordance with your comments, we have made the following adjustments to the article:

  1. We have supplemented the introduction section with all the necessary sources to support our statements.
  2. We have supplemented the introduction section with links to all sources of statistical information. These sources are extremely reliable, their description was also added in the introduction.
  3. Thank you very much for your indication of the need to describe Figure 1. We have detailed this description.
  4. We have decoded all abbreviations (SES, WES, BIOEP and others) and corrected the spelling of CO2
  5. Thus, the primary stage of this study is aimed at identifying the most perspective NCRES influencing traditional energy system. Effect will be more significant in developing energy systems, which start enrgy transition to renewables, so it is possible to use Russian energy system as an example to prove the model. That is why the model was created based on statistical information for the Russian Federation. However, the architecture of the resulting model is relatively universal and the conclusions are significant, which makes it possible to extrapolate this architecture to other developing countries. Moreover, the conclusions made based on this model formed the basis for the analysis of the information environment within the framework of the "Discussion" section. The conclusions there are universal and relevant at the global level. We tried to describe this specificity in the Methodology section.
  6. We tried to detail the information on the model shown in Figure 2. The graph has a characteristic non-linearity, which does not allow using simple polynomial time trends to assess the development of NCRES. Usage of such trend models can lead to a poor result shown in Table 1. Because of that we suggest developing a more advanced model which will be described below.
  7. Regarding the use of the time trend model, we can say that Usage of the trend models is the most common way to predict time series. Some advanced model in renewables sphere were described in article [Sang-Bing Tsai, Youzhi Xue, Jianyu Zhang, Quan Chen, Yubin Liu, Jie Zhou, Weiwei Dong, Models for forecasting growth trends in renewable energy, Renewable and Sustainable Energy Reviews, Volume 77, 2017, Pages 1169-1178, ISSN 1364-0321]. These models are easy to create because they don’t need to have any additional data except data that is going to be predicted.
  8. It can be noted that for solar power plants prediction is in significance interval and it still can be used for a prediction modeling and forecasting, but have lowest accuracy across prediction of NRCES that is caused by significant non-linearity in data.
  9. Thank you so much for pointing out the error regarding regional division in the Russian Federation. This is our mistake of translation into English, because of it the meaning of the sentence has completely changed. We have corrected it. The presented model does not extensively highlight the specific sectors of renewable energy sources prevailing in each region of Russian Federation, focusing more on the rate of development of these technologies on the whole.
  10. Concerning the factors presented, Scientific potential allows us to estimate how many really important technologies are being implemented in projects. Industry growth assessments reflect projects that directly impact the industry. Discovery potential is needed to assess the emergence of new technologies in the industry, which will lead to higher growth rates.
  11. Regarding the terms in Figure 6 - thank you very much for pointing out this defect, we tried to reveal them in Table 2.
  12. To make more precise prediction we estimated the regional potential of development of NCRES in Russia and integrated them with the model.

We have tried to correct the article, focusing on the scientific significance of the study, and we very much hope for your favor.

Round 2

Reviewer 1 Report

The manuscript has been corrected but point II.1 from the first round have to be taken into account. 

II. Comments to the Materials and Methods, Results and Discussion sections:

  1. I suggest rewriting the Materials and Methods, Results and Discussion sections according to the journal's Instructions for Authors. Currently, the information in these chapters is mixed. For example information about the algorithm (line 235) and mathematical model (line 256), the paragraph in line 411 or 420, etc. The current layout of the content may make the article difficult for the reader to understand.

Author Response

Dear reviewer, thank you very much for this comment. We have completely redesigned the Materials and Methods and Results sections and slightly improved the Discussion section. At the moment, the Materials and Methods section fully describes the methodology of this study, and the Results section concentrates on the final result of applying the presented methodology. We hope very much for your favor. Thank you!

Reviewer 2 Report

I think the authors for their efforts to improve the quality of the manuscript. However, the problem with scientific contribution of this paper remains the same. The addressed prediction model really uses the traditional regression methodology without any critical analysis or comparative models.

In my opinion, this paper may be interesting for an editorial or a communication paper. The scientific contribution of the work is not enough significant to be considered for a regular research paper.

Author Response

Dear reviewer, thank you very much for this comment. We have completely redesigned the Materials and Methods and Results sections and slightly improved the Discussion section. At the moment, the Materials and Methods section completely describes the methodology of this study, and the Results section concentrates on the final result of applying the presented methodology. The results are not only the model itself, but the systemic conclusions drawn on its basis, as well as the results of the analysis of the information environment. We hope very much for your favor. Thank you!
